# Effect of Different Tool Probe Profiles on Material Flow of Al–Mg–Cu Alloy Joined by Friction Stir Welding

**DOI:** 10.3390/ma14216296

**Published:** 2021-10-22

**Authors:** Anton Naumov, Evgenii Rylkov, Pavel Polyakov, Fedor Isupov, Andrey Rudskoy, Jong-Ning Aoh, Anatoly Popovich, Oleg Panchenko

**Affiliations:** 1Institute of Mechanical Engineering, Materials and Transport, Peter the Great St. Petersburg Polytechnic University, 195251 St. Petersburg, Russia; rylkov_en@spbstu.ru (E.R.); polyakov_py@spbstu.ru (P.P.); isupov_fyu@spbstu.ru (F.I.); rector@spbstu.ru (A.R.); director@immet.spbstu.ru (A.P.); panchenko_ov@spbstu.ru (O.P.); 2Advanced Institute of Manufacturing with High-Tech Innovations (AIM-HI), National Chung Cheng University, Min-Hsiung, Chiayi 621, Taiwan; imejna@ccu.edu.tw

**Keywords:** Al–Mg–Cu alloy, friction stir welding (FSW), probe profile, microstructure evolution, mechanical properties, material flow, CFD modeling

## Abstract

Friction Stir Welding (FSW) was utilized to butt−join 2024–T4 aluminum alloy plates of 1.9 mm thickness, using tools with conical and tapered hexagonal probe profiles. The characteristic effects of FSW using tools with tapered hexagonal probe profiles include an increase in the heat input and a significant modification of material flow, which have a positive effect on the metallurgical characteristics and mechanical performance of the weld. The differences in mechanical properties were interpreted through macrostructural changes and mechanical properties of the welded joints, which were supported by numerical simulation results on temperature distribution and material flow. The material flow resulting from the tapered hexagonal probe was more complicated than that of the conical probe. If in the first case, the dynamic viscosity and strain rate are homogeneously distributed around the probe, but in the case of the tapered hexagonal probe tool, the zones with maximum values of strain rates and minimum values of dynamic viscosity are located along the six tapered edges of the probe.

## 1. Introduction

Aluminum alloys of Al–Cu series (2xxx) belong to the wrought age−hardening aluminum alloys, and are widely used in aircraft and automotive structures and in the oil and gas industry for the manufacturing of structural components [1,2,3,4]. However, the range of applications is limited by the comparatively low corrosion resistance and weldability.

Since Al–Cu alloys contain more alloying elements and impurities, the types of intermetallic phases formed in the alloys are likely to be the most diverse among aluminum alloys. Dispersion hardening of Al–Cu alloys occurs due to the precipitation from the solid solution and growth of the equilibrium θ−phase Al_2_Cu, as well as intermediate phases and clusters. By adding Mg and Mn to the Al–Cu alloys to increase their strength [5], a hardening effect is achieved due to the precipitation and growth of more complex intermetallic phases. Depending on the ratio of the Cu and Mg concentrations, different equilibrium phases may be produced. When the ratio of Cu:Mg is greater than 8:1, the precipitation hardening is mainly caused by the θ−phase of Al_2_Cu. However, when the ratio of Cu:Mg is between 8:1 and 4:1, the precipitation hardening is caused by the θ−phase of Al_2_Cu and the S−phase of Al_2_CuMg. When the Cu:Mg ratio is between 4:1 and 1.5:1, the S−phase of Al_2_CuMg becomes the main hardening phase [6]. The last−mentioned ratio of Cu:Mg is applicable to the Al–Cu–Mg alloy AA 2024. The addition of a small amount of silicon into the Al–Cu–Mg alloys promotes the formation of the β−phase of Mg_2_Si, which has a positive effect on the kinetics of artificial aging [6].

Until the invention and increasing implementation of the solid−state welding technology—friction stir welding (FSW)—in 1991 [7], components made of Al–Cu alloys were usually joined using the rivets process. At present, due to a number of advantages over fusion welding, FSW technology is being used successfully and increasingly for welding components made of age−hardening aluminum alloys. When using FSW, there are no hot cracks and porosity in the weld; the residual stresses are significantly reduced, thereby minimizing the distortion; and the strength characteristics of the weld are improved due to grain refinement in the stir zone (SZ). However, with age−hardening alloys there is still the same problem with FSW, as is the case with other fusion welding processes, of a strong reduction in strength in the transition zones of the weld, namely the thermomechanical affected zone (TMAZ) and the heat affected zone (HAZ). At elevated temperatures, the particles of the hardening phase increase in size or dissolve, which makes the HAZ the weakest area [8]. The fracture of friction stir−welded joints of Al–Cu–Mg alloys occurs along the boundary between SZ and TMAZ due to the presence of larger particles of the Al_2_Cu phase along the grain boundaries [9,10,11].

Friction stir welding on thin sheets is preferable to fusion welding because of its lower heat generation and thus minimization of the distortion of the welded sheets [12,13,14], however, friction stir welding on thin sheets could be complicated due to the presence of remnant oxide line (ROL) in the weld. Not only the length of the probe, but also other additional features, such as type of thread or pattern of probe, are required to be modified to break the oxide line in the SZ. Therefore, variations of probe profile and geometry, such as triangular (TR), square (SQ), pentagon (PEN), and hexagon (HEX) have been used to improve the quality of the joint by intensifying the stir and breaking of the oxide layer in SZ [15,16,17,18,19,20,21,22,23,24].

The purpose of the present research is to investigate the effect of tapered hexagonal tool probe profile on the macrostructure, mechanical performance and material flow of AA 2024−T4 joined by friction stir welding.

## 2. Materials and Methods

### 2.1. Base Metal

In this study, heat−treatable AA 2024 (Al–Mg–Cu) alloy sheets of 1.9 mm thick after hot rolling and subsequent natural aging (T4) were used. The chemical composition of the AA2024 alloy is listed in Table 1.

The optical microstructure of the base material is presented in Figure 1a,b. The grain structure of the wrought base material was dominated by equiaxed grains with an average grain size of 18 µm. The tensile properties of the base material are as follows: UTS = 450 MPa; YS = 310 MPa; El = 13%.

### 2.2. Friction Stir Welding Procedure

The sheets with dimensions of 400 mm × 100 mm × 1.9 mm were butt−welded parallel to the rolling direction using FSW technique. Two different tools were used in this research: 1—a tool with a conical probe composed of a smooth flat shoulder of 10 mm in diameter and a smooth conical probe of 3.8 mm in diameter at the shoulder end and 3.0 mm in diameter at free end, as depicted in Figure 2a; 2—a tool with a tapered hexagonal probe composed of a smooth flat shoulder of 10 mm in diameter and a tapered hexagonal probe with a diagonal length of 3.8 mm at the shoulder end and 3.0 mm at the free end of the probe, as shown in Figure 2b. The length of both probes is 1.8 mm. The tilt angle was 2°, the axial force in vertical direction at steady friction stir stage was 4 kN. Friction stir welding was performed on Matec 40P gantry machine (Matec GmbH, Kongen, Germany). Welding conditions and process parameters are listed in Table 2. The plunging and dwelling stages of the FSW process were guaranteed by position control mode, the traverse stage of FSW process was guaranteed by force control mode. The parameters were selected according to literature review [25,26,27] and according to the previous research [2,28].

The weld pitch WP (mm/rpm) factor was determined as the relationship between welding speed and rotation speed, Table 2.

The total heat input Q_total_ (Watt) was calculated using the contact shear stress, τ_contact_ (Pa), tool angular rotation speed, ω (rad/s), shoulder radius, R_shoulder_ (mm), probe radius, R_probe_ (mm) and probe length, R_probe_ (mm) by Equation (1):Q_total_ = 2/3π τ_contact_ ω (R^3^_shoulder_ + 3R^2^_probe_H_probe_),(1)

The τ_contact_ and ω were determined by axial force F_z_ (N) and rotation speed N (rpm) under constant friction coefficient µ = 0.4 [29,30,31] using Equations (2) and (3), respectively:τ_contact_ = µF_z_/πR^2^_shoulder_,(2)
ω = 2π N/60,(3)

The linear energy, l (J/mm) was calculated as a relation between the heat input and welding speed υ (mm/s). The difference in the heat input as well as linear energy is highlighted in Table 2. Friction stir welding with tapered hexagonal probe (FSW−H) exhibited higher total heat input and linear energy than with conical probe (FSW−C).

### 2.3. Macrostructural and Mechanical Properties Analysis

Samples for optical microscopy and hardness testing were cut perpendicular to the welding direction in accordance with Figure 3 and prepared by standard metallographic preparation techniques. The polished samples were etched with reagent consisting of 2 mL HNO_3_, 5 mL CH₃COOH, 12 mL H_3_PO_4_ and 1 mL H_2_O for 20 s at a temperature of 60 °C. Vickers microhardness was measured along the two lines at the distances 0.6 and 1.2 mm from the bottom surface of the transverse weld section with a load of 0.98 N (HV 0,1) for 10 s after several weeks. Tensile tests were carried out on a universal machine Zwick/Roell Z100 (Zwick/Roell GmbH, Ulm, Germany) with crosshead speed of 10 mm/min at room temperature according to Russian Standard (GOST 6996). No thickness reduction in the joints after FSW was found. The mechanical properties of the weld (ultimate tensile strength and yield stress) were determined by testing five replicates for each weld condition.

### 2.4. Numerical Simulation

Computational Fluid Dynamics (CFD) modelling was used to determine the temperature and effective strain rate distribution during FSW. The computational domain of the FSW model, tool position, and tool geometry are presented in Figure 4.

It should be noted that the tool shoulder surface was only party in contact with the workpiece. Material flows into the computational domain through the ‘Inlet’ and ‘Outlet’ boundaries with a speed of 300 mm/min identical to the welding speed applied in the experiments. The speeds of the ‘Sides’, ‘Top surface’, and ‘Down surface’ were also set as 300 mm/min. The convection heat transfer coefficient was assumed to be 300 W/m^2^K at the ‘Down surface’ and 30 W/m^2^K at other surfaces. The surface that was in contact with the tool was rotated about the Y axis with the speed according to the parameters in Table 2. The CFD model contains 8,637,083 tetrahedral cells with finer mesh in the center. The governing equations were described in previous research [32,33,34].

The validation of the CFD model was provided experimentally by means of temperature measurement using a thermocouple welded to the top surface of the plates on both sides (AS and RS) of the weld at a distance of 7 mm from the center line according to the scheme illustrated in Figure 3. The measured and calculated thermocycles are shown in Figure 5, and the calculated values are in good agreement with the measured ones.

## 3. Results and Discussion

### 3.1. Macrostructural Characteristics

Figure 6 shows the typical cross−sectional macrostructures of the friction stir welds which were produced using tools with conical (Figure 6a) and tapered hexagonal (Figure 6b) probes. The continuous thin zigzag line in the weld center extending from the top surface to the bottom was revealed in the etched macrographs of both of the examined samples, Figure 6a,b. This appeared to be a remnant oxide layer (ROL) which represents the oxide films (e.g., Al_2_O_3_) from the initial butt surfaces broken by the probe during the stirring process [35,36,37]. The ROL in the macrostructure of the sample welded with the conical probe (Figure 6a) exhibits different features in contrast to that welded with the tapered hexagonal probe (Figure 6b). In the case of using the conical probe, ROL passes through the top to the bottom of the entire nugget zone and reveals a curved long loop toward the retreating side (RS) in the lower part of the joint. While in case of using the tapered hexagonal probe, ROL passes through the top to the bottom of the nugget zone without revealing a curved loop. The conical and tapered hexagonal probes obviously resulted in different material flows.

In the macrostructures of the friction stir welds, three zones are generally defined. The stir zone (SZ) is located in the weld center and is characterized by a fine−grain recrystallized microstructure. The thermomechanically affected zone (TMAZ) is composed of highly deformed grains. The heat−affected zone (HAZ) is where the structure was influenced only by temperature and the grain structure was almost the same as that of the unaffected base metal (BM). As can be seen from the macrostructures in Figure 6, the contour of the zones varied as tool probes with different geometries were applied. The SZ of the weld achieved by the tapered hexagonal probe shows a narrower width due to the probe’s geometries. The weld produced by the tapered hexagonal probe exhibits a wider HAZ than that produced by the conical probe.

### 3.2. Microhardness and Tensile Behaviour

The microhardness distributions along the transverse direction of the welds are presented in Figure 7. The microhardness of the base metal was approximately 120 HV. The microhardness distribution of all welds depicts a W−shaped profile, with a wide, softened HAZ which is typical of the friction stir weld of heat−treatable aluminum alloys. The hardness distribution in the HAZ toward the retreating side (RS) reveals lower values, as shown in Figure 7a,b, where the minimum hardness level is approximately 105 HV. As can be seen, the hardness level in the SZ produced by the conical probe is lower (about 115 HV in average) than that produced by the tapered hexagonal probe (about 120 HV in average). The sample FSW−H shows slightly higher hardness in the SZ than FSW−C. A narrower SZ closer to the bottom of the joints is a feature of the macrostructure of the FSW joints, and can be seen in Figure 7a,b. At a distance of 0.6 mm from the bottom (black lines), hardness decreases in the HAZ at a distance of approximately 2 mm from the weld center. At a distance of 1.2 mm from the bottom (blue lines), hardness decreases in the HAZ at a distance of approximately 4 mm from the weld center on both AS and RS for the FSW−H sample (Figure 7a) and on RS for the FSW−C sample (Figure 7b). The increase in the hardness in the SZ on the AS could be explained by the intensifying of the phenomena of reprecipitation at higher temperatures compared with the RS, Figure 8.

The average tensile properties of the studied joints, out of a total of five tensile specimens, were determined and are listed in Table 3. The mechanical properties of the weakest section of a tensile specimen are generally accepted as the global properties of the joint. In general, the fracture of the tensile specimen taken from the transverse section of a friction stir weld of heat−treatable aluminum alloys occurs in the HAZ, assuming a defect−free weld [8]. As described in the scientific literature [38,39], the existence of weakest region was due to dissolution and coarsening of the strengthening precipitates resulting from welding heat, and rendered a decrease in mechanical properties. It is worth noting that all FSW tensile specimens exhibited their fracture locations in the SZ.

### 3.3. Temperature Fields

The CFD model was used to analyze the temperature field and material flow during FSW. Calculated temperature distributions at steady friction stir stage for both FSW−C and FSW−H samples are presented in Figure 8 and Figure 9. The temperature distribution of the FSW−C sample exhibits a region of maximum temperature only located under the probe, whereas the temperature distribution of the FSW−H sample reveals a larger region of maximum temperature, which extended from the probe tip to the adjacent area towards the advancing side. In both cases, the temperature fields depict a nonsymmetric distribution about the probe axis, with higher temperatures towards the advancing side. The difference between the maximum calculated temperatures of FSW−H and FSW−C is less than 15 °C. However, FSW−H shows a slightly larger volume of metal in the SZ that is heated to maximum temperature.

To better understand the effect of the tapered hexagonal probe on the joint formation, dynamic viscosity and strain−rate distributions were calculated. The distributions of the dynamic viscosity in the transverse section and in the longitudinal section of the joints are presented in Figure 10a,b, respectively. The distributions of the strain rate in the transverse section and in the longitudinal section of the joints are presented in Figure 11a,b, respectively.

In case of the conical probe and in contrast to the case of the tapered hexagonal probe, a small circular rim of higher dynamic viscosity is located at the root of the conical probe, directly adjacent to the shoulder as depicted in Figure 10 (left). The higher the dynamic viscosity, the more difficult the material is to be stirred, therefore the strain−rate distribution corresponding to the small, circular rim area of higher viscosity exhibits a lower value of strain rate, as depicted in Figure 11 (left). It is also noticeable that strain rates for FSW−H sample in the transverse section are higher than those in the longitudinal section (Figure 11a). To emphasize the influence of the probe profile on the material flow, the distribution of the dynamic viscosity and strain rate in the plane of the welded sheets at a distance of half of the thickness was calculated and presented in Figure 12 and Figure 13.

It can clearly be seen that in the case of tapered hexagonal probe tool, the material flow is somewhat complicated compared with that of the cylindrical probe tool. In the case of the FSW−C sample, the dynamic viscosity and strain rate are homogeneously distributed around the probe, while in the case of the FSW−H sample zones, it is revealed that maximum values of strain rates and minimum values of dynamic viscosity are concentrated along the six tapered edges.

The difference in material flow for the two probe profiles within the stir zone could be illustrated by the isosurfaces of flow stress of 200 MPa, as shown in Figure 14. The iso−surfaces look similar for both probe profiles directly under the shoulder and close to the outer edge. However, the flow stress distribution around the two tool probes reveals different morphologies. It can be clearly seen that near all edges of the tapered hexagonal probe, the material reaches the stress of 200 MPa because of high values of strain rate in these zones compared with those of the conical probe, where flow stress of 200 MPa could be found only under and near the bottom tip of the probe.

It is difficult to calculate the amount of strain with CFD models. Yet, it is possible to calculate the strain characteristic, known as “pseudo strain”. The CFD model used in this study has a stationary tool, which only rotates without translation along the weld line. The material moves towards the tool with the welding speed (as in this study the welding speed is 300 mm/min for both tool geometries). We assume that the material is subjected to plastic deformation when dynamic viscosity is lower than 5 × 10^6^ Pa∙s [40]. This zone, wherein the dynamic viscosity is lower than 5 × 10^6^ Pa∙s, is limited by the green circle shown in Figure 15. In the absence of deformation during welding, the material would move and intersect with this area in a straight line from the yellow to black marker, and the distance between these markers is L_0_. In the case of FSW with deformation, the material moves along the streamlines, shown in Figure 15 as blue lines, and its length is L_1_. Thus, it is possible to calculate “pseudo strain” as the ratio of (L_1_ − L_0_) and L_0_, as described in Equation (4).
pseudo strain = (L_1_ − L_0_)/L_0_,(4)

“Pseudo strain” values were calculated for three points in the middle plane at distances of 0 mm, 1 mm and 1.5 mm from the center line in the longitudinal direction of the weld for both types of tool geometry. The results are presented at Table 4.

The comparison of the results presented in Table 4, with the assumptions mentioned, allows us to conclude that “pseudo strains” for FSW−C and FSW−H are close to each other, but the FSW−C approach has higher values than FSW−H. The closer the streamlines are to the pin, the higher the difference in “pseudo strain” between FSW−C and FSW−H.

It was observed that dynamic recrystallization occurred in aluminum when the amount of strain was larger than 0.2, at a temperature of 300 °C and strain rate of 10 s^−1^ [41]. In the SZ of both cases, the temperature is higher than 350 °C, and the “pseudo strains” are larger than 0.3. This allows us to assume that dynamic recrystallization proceeds in SZ. Thus, the microstructure in the SZ is determined by dynamic recrystallization since the conical or tapered hexagonal tool probe profile, being a heat source, affects the material flow and therefore the shape of SZ. Furthermore, the tool profile also has a relevant effect on mechanical stirring and breaking of the remnant oxide layer.

## 4. Conclusions

Al–Mg–Cu aluminum alloy plates were friction stir−welded with conical and tapered hexagonal tool probe profiles, in order to investigate the influence of the tapered hexagonal probe profile on the material flow, macrostructure and mechanical performance of the joint. The results obtained in the present study are summarized as follows:(1)In the case of the conical probe, ROL passes through the top to the bottom of the entire nugget zone, and reveals a curved, long loop toward the retreating side (RS) in the lower part of the joint. While in the case of the tapered hexagonal probe, ROL passes through the top to the bottom of the nugget zone, without revealing a curved loop. The conical and tapered hexagonal probes obviously resulted in different material flows.(2)A narrower SZ closer to the bottom of the joint is a feature of the macrostructure of the weld produced by the tapered hexagonal probe. The SZ of the weld produced by the tapered hexagonal probe reveals a narrower width than that of the conical probe, while the weld of the tapered hexagonal probe reveals a wider HAZ than that of the conical probe. This macroscopic observation was supported by calculation results.(3)The average ultimate tensile strength of the welds of FSW−H specimens reached a higher value of 354 MPa compared with that achieved by the FSW−C probe, of 288 MPa. This could be attributed to the more intensive material flow around the probe. Yielding strength is almost the same for both probe types.(4)In the case of the FSW−C sample, the dynamic viscosity and strain rate are homogeneously distributed around the probe, while in the case of the FSW−H sample, maximum strain rate and minimum dynamic viscosity are located along the six tapered edges.(5)The temperature distribution of the FSW−C sample exhibits a region of maximum temperature located beneath the probe end, whereas the temperature distribution of the FSW−H sample reveals a larger region of maximum temperature which extends from the probe tip to the adjacent area towards the advancing side.(6)The probe profile has a relevant effect on the stirring and breaking of the remnant oxide layer. The microstructure evolution in the SZ was dominated by dynamic recrystallization.

## Figures and Tables

**Figure 1 materials-14-06296-f001:**
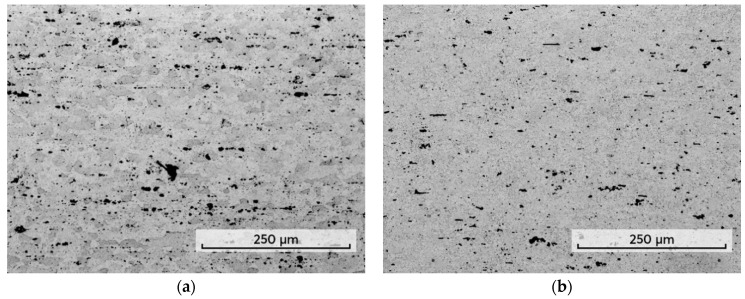
Microstructures of base material: (**a**) optical image transverse to the rolling direction; (**b**) optical image in longitudinal direction (rolling direction).

**Figure 2 materials-14-06296-f002:**
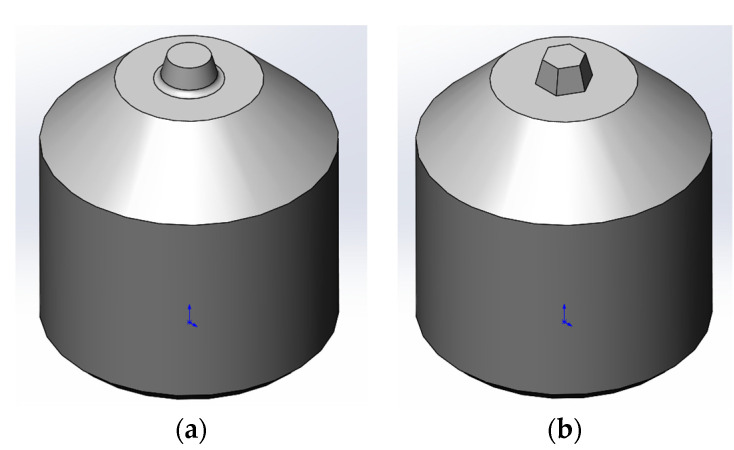
Tools for FSW with different probe profiles: (**a**) conical form; (**b**) tapered hexagonal form.

**Figure 3 materials-14-06296-f003:**
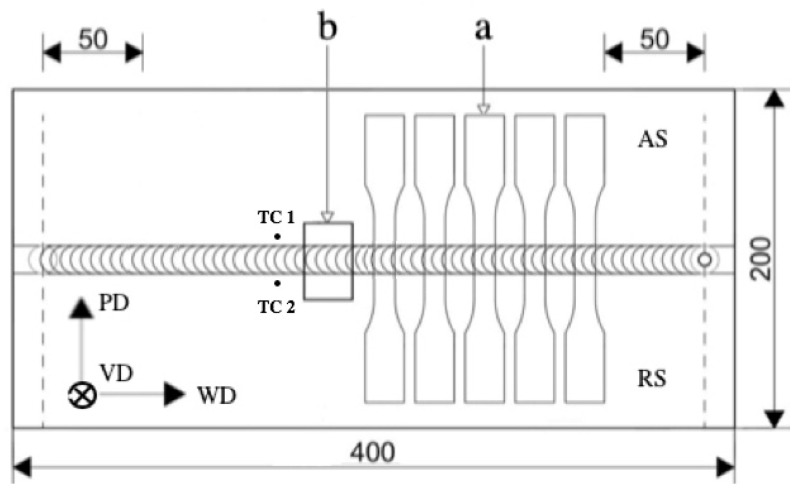
Locations of the specimens extracted for tensile testing (a); specimen for optical microscopy and microhardness measurement (b); TC 1 and TC 2—thermocouples for temperature measurement; WD—welding direction, PD—transverse direction, VD—vertical direction; the dimensions are mm in unit.

**Figure 4 materials-14-06296-f004:**
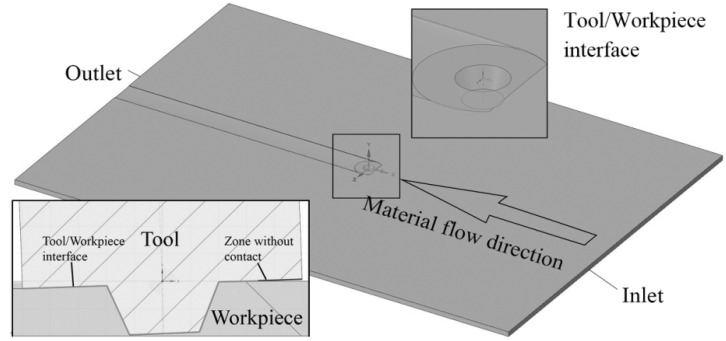
Scheme of the domain, tool position, and tool geometry.

**Figure 5 materials-14-06296-f005:**
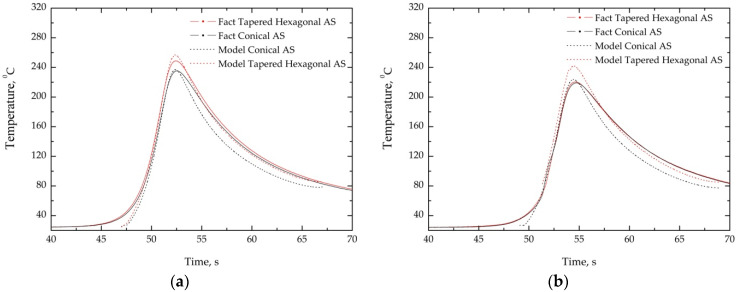
The measured and calculated thermocycles during FSW using cylindrical and tapered hexagonal probe on (**a**) advancing side (AS); and on (**b**) retreating side (RS).

**Figure 6 materials-14-06296-f006:**
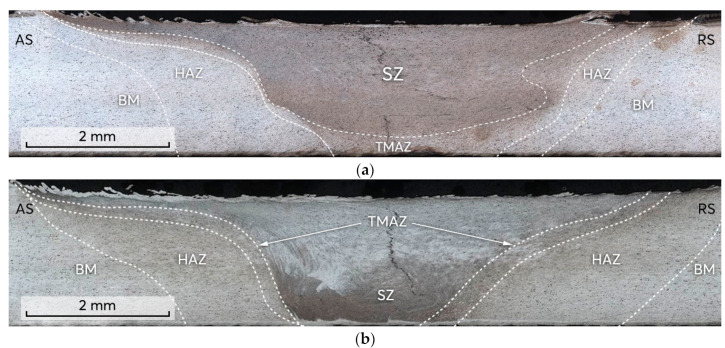
Transverse macrostructure sections of: (**a**) FSW−C, (**b**) FSW−H joints revealing remnant oxide layer (ROL).

**Figure 7 materials-14-06296-f007:**
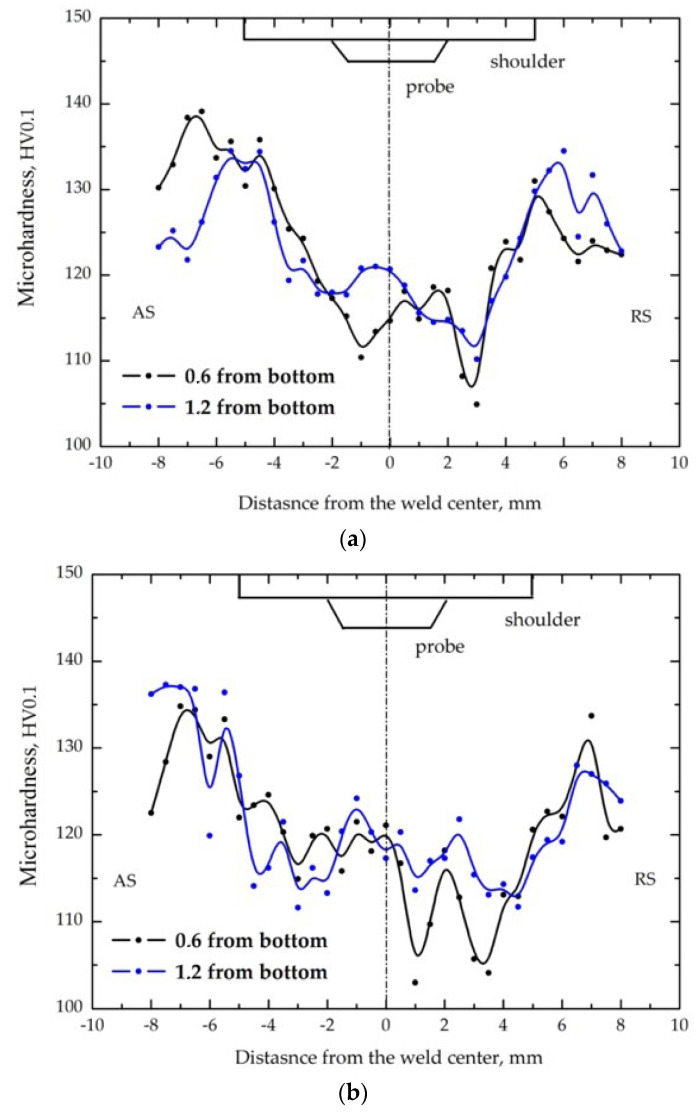
Microhardness profile of the studied joints: (**a**) FSW−C and (**b**) FSW−H.

**Figure 8 materials-14-06296-f008:**
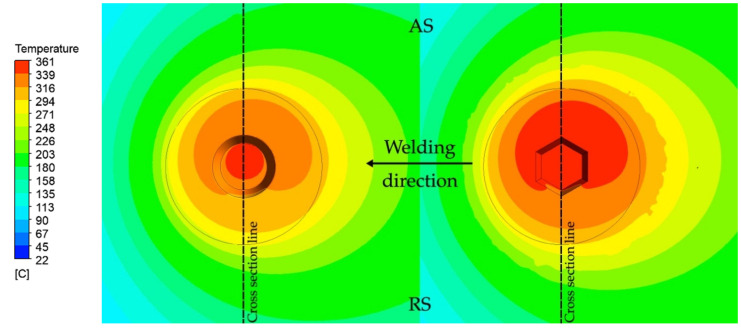
Temperature distribution for both FSW−C (**left**) and FSW−H (**right**), top view.

**Figure 9 materials-14-06296-f009:**
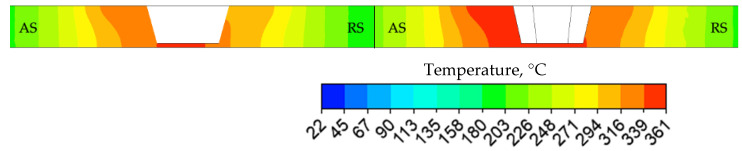
Temperature distribution for both FSW−C (**left**) and FSW−H (**right**), transverse section (according to the cross−section line in Figure 8).

**Figure 10 materials-14-06296-f010:**
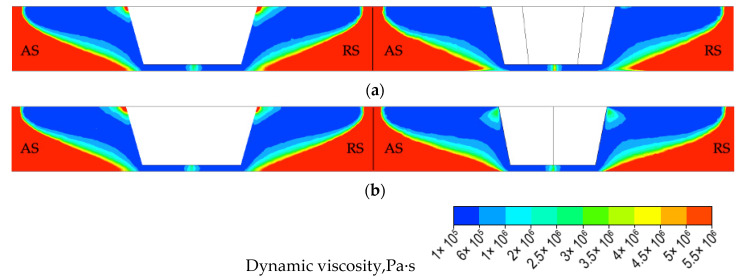
The distribution of the dynamic viscosity for both FSW−C (**left**) and FSW−H (**right**) samples, (**a**) transverse and (**b**) longitudinal section.

**Figure 11 materials-14-06296-f011:**
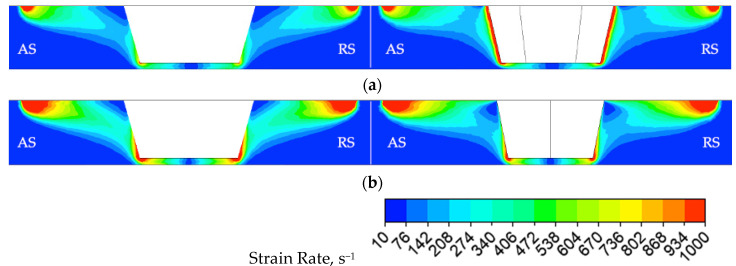
The distribution of the strain rate for both FSW−C (**left**) and FSW−H (**right**) samples: (**a**) transverse and (**b**) longitudinal sections.

**Figure 12 materials-14-06296-f012:**
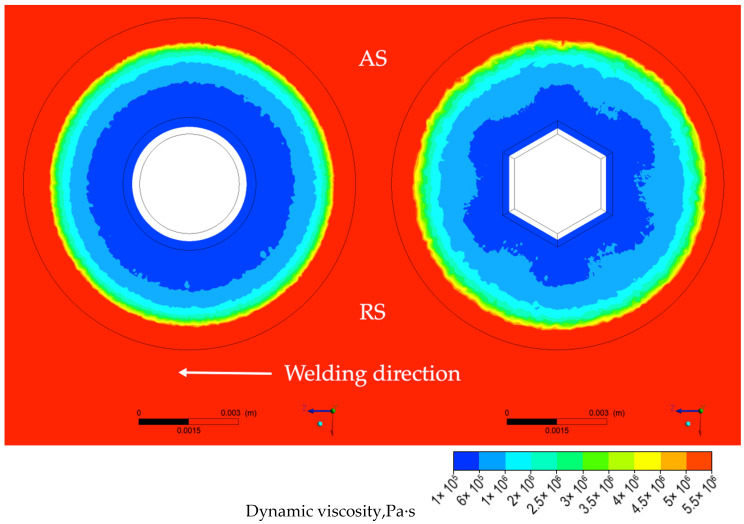
The distribution of the dynamic viscosity and strain rate in the plane of the welded sheets at a distance of half of the thickness for both samples FSW−C (**left**) and FSW−H (**right**).

**Figure 13 materials-14-06296-f013:**
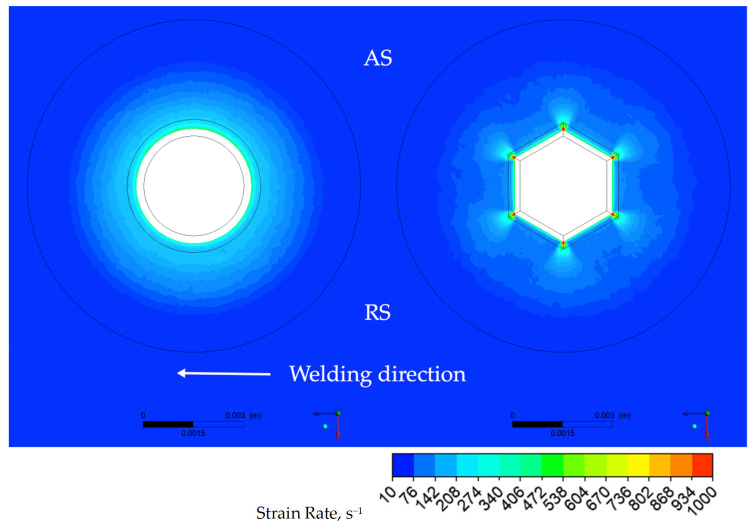
The distribution of the strain rate in the plane of the welded sheets at a distance of half of the thickness for both samples, FSW−C (**left**) and FSW−H (**right**).

**Figure 14 materials-14-06296-f014:**
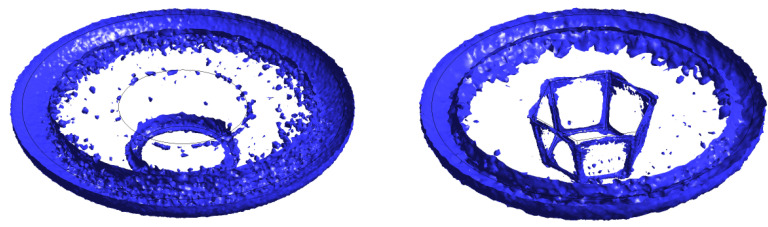
Isosurface of flow stress equal to 200 MPa for both FSW−C (**left**) and FSW−H (**right**) samples.

**Figure 15 materials-14-06296-f015:**
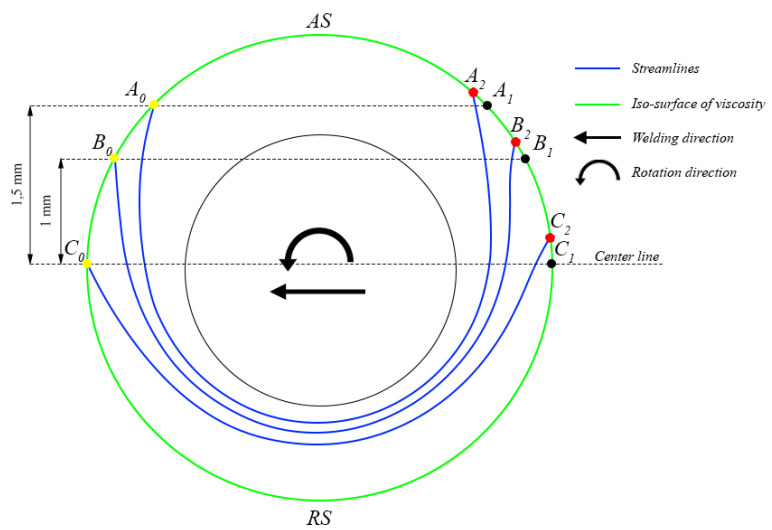
Scheme of “pseudo strain” calculation for the cylindrical probe profile.

**Table 1 materials-14-06296-t001:** Chemical composition of Aluminum Alloy 2024 (wt%) used in the study.

Fe	Si	Cu	Mn	Mg	Zn	Al
0.25	0.11	3.80	0.41	1.30	0.22	Bal.

**Table 2 materials-14-06296-t002:** FSW matrix.

Weld ID	Probe Form	N, rpm	υ, mm/min	WP, mm/rpm	Q_total_, Watt	l, J/mm
FSW−C	Conical	1200	300	0.250	728	146
FSW−H	Tapered Hexagonal	1200	300	0.250	754	151

**Table 3 materials-14-06296-t003:** Tensile properties of the studied joints.

Specimen	UTS (MPa)	YS (MPa)
FSW−C	288.3	275.9
FSW−H	354.8	279.2

**Table 4 materials-14-06296-t004:** “Pseudo strain” values.

Start Point Position	FSW−C	FSW−H
0 mm	0.385	0.366
1 mm	0.892	0.814
1.5 mm	2.493	2.269

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
