# Peer review of "Effect of Different Tool Probe Profiles on Material Flow of Al–Mg–Cu Alloy Joined by Friction Stir Welding"

_materials, 2021, doi:10.3390/ma14216296_

Round 1
Reviewer 1 Report
This paper studies the friction stir welding ( FSW ) method for butt joining aluminum alloy plates with the age-hardening aluminum alloy. The main contribution of the paper is to analyze the differences in mechanical properties of the welded joints, supported by the results of the numerical simulation of the temperature distribution and material flow. The paper is well-written, but, I have some concerns about the technical contribution of this paper:
- The novelty of this paper is limited. The influence of the sony geometry in the FWS welding process is investigated and the results are published. Authors should focus more on novelty and their own contribution to research against the background of existing publications (For example, Wronska et al. Effect of tool pin length on microstructure and mechanical strength of the FSW joints of Al 7075 metal sheets).
- What are the limitations of the adopted research method and what are the planned further directions of research?
- It should expand the discussion and application sections for information on the applicability of the results in practice. In my opinion, there is no specific information in the conclusions as to what the probe geometry guarantees that better parameters will be obtained.
Author Response
Thank you very much for reviewing the article “Effect of different tool probe profiles on material flow of Al-Mg-Cu alloy joined by friction stir welding”.
Remark 1) - The novelty of this paper is limited. The influence of the sony geometry in the FWS welding process is investigated and the results are published. Authors should focus more on novelty and their own contribution to research against the background of existing publications (For example, Wronska et al. Effect of tool pin length on microstructure and mechanical strength of the FSW joints of Al 7075 metal sheets).
The novelty of the paper is in the implementation of CFD modeling method to the simulation of material flow enhanced by the different profile (not the length or diameter) of the probe in the horizontal plane – round or hexagonal. Moreover, the method of pseudo-strain calculation is suggested. The text of the paper was revised significantly.
Remark 2) - What are the limitations of the adopted research method and what are the planned further directions of research?
The limitation of CFD method is the lack of the strain distribution calculation. That is why the method of pseudo-strain calculation is suggested. The further direction of the research is to improve more the material stirring by means of different probe profiles and additional features. The text of the paper was revised significantly.
Remark 3) - It should expand the discussion and application sections for information on the applicability of the results in practice. In my opinion, there is no specific information in the conclusions as to what the probe geometry guarantees that better parameters will be obtained.
The corrections to the conclusions have been made. The text of the paper was revised significantly.

Reviewer 2 Report
A very interesting article.
Author Response
Thank you very much for reviewing the article “Effect of different tool probe profiles on material flow of Al-Mg-Cu alloy joined by friction stir welding”.
Thanks for your review. We fixed errors in the article, improved the quality of the English language, added a detailed description of the methodology.

Reviewer 3 Report
This paper contains the useful information in the engineering field.
Please recheck the following items.
1) line 91: Figure 2b,c ?
2) Please discuss the relation among the hardness distribution (Figure 7), temperature distribution (Figure 8) and the strain rate distribution (Figure 11).
3) line273:” the green circle shown in Figure 5”?
4) line276: “shown at Figure 5”?
Author Response
Thank you very much for reviewing the article “Effect of different tool probe profiles on material flow of Al-Mg-Cu alloy joined by friction stir welding”.
Remark 1) “line 91: Figure 2b,c ?”
The corrections have been made.
Remark 2) “Please discuss the relation among the hardness distribution (Figure 7), temperature distribution (Figure 8) and the strain rate distribution (Figure 11)”
Figure 8 shows that the temperature on the AS is higher than on the RS. In the higher temperature zone, the microhardness of the material is higher due to the possible reprecipitation process, that is shown in Figure 7. In case of FSW-H, on the AS there is the highest temperature and the highest microhardness.
Discussion has been added to the article.
Remark 3) ”the green circle shown in Figure 5”
The corrections have been made.
Remark 4) “shown at Figure 5”
The corrections have been made.

Reviewer 4 Report
The manuscript studies the effect of tapered hexagonal tool probe profile on macrostructure, mechanical performance and material flow of AA 2024 -T4 joined by friction stir welding. Through the combination of simulation calculation and experiments, the research is logical and clear, but there are still some problems to be modified:
- The references cited in this manuscript are very old, and more studies in the past five years are needed to prove the development of the research.
- The introduction part of the paper is insufficient and needs to be enriched. Here are several literatures about molten channel forming for reference only.
- The format rigor of the manuscript needs to be improved. For example, figure 2c in line 91 does not exist.
Author Response
Thank you very much for reviewing the article “Effect of different tool probe profiles on material flow of Al-Mg-Cu alloy joined by friction stir welding”.
Remark 1) “The references cited in this manuscript are very old, and more studies in the past five years are needed to prove the development of the research.”
The text of the paper was revised significantly. The relatively old references were replaced by “fresh” ones.
Remark 2) “The introduction part of the paper is insufficient and needs to be enriched. Here are several literatures about molten channel forming for reference only”
The text of the paper was revised significantly. New references were added to the paper.
Remark 3)”The format rigor of the manuscript needs to be improved. For example, figure 2c in line 91 does not exist”
The corrections have been made.
